# Agronomic Efficiency of Animal-Derived Organic Fertilizers and Their Effects on Biology and Fertility of Soil: A Review

Shantanu Bhunia [1] , Ankita Bhowmik [1], Rambilash Mallick [2] and Joydeep Mukherjee [1,*]

1   School of Environmental Studies, Jadavpur University, Kolkata 700032, India; shanu.bhunia@gmail.com (S.B.); ankitabh30@gmail.com (A.B.)
2   Department of Agronomy, Institute of Agricultural Science, University of Calcutta, Kolkata 700019, India; rbmallick@rediffmail.com
*   Correspondence: joydeep.mukherjee@jadavpuruniversity.in; Tel.: +91-33-2414-6147; Fax: +91-33-2414-6414

**Abstract:** Healthy soils are essential for progressive agronomic activities. Organic fertilization positively affects agro-ecosystems by stimulating plant growth, enhancing crop productivity and fruit quality and improving soil fertility. Soil health and food security are the key elements of Organic Agriculture 3.0. Landfilling and/or open-dumping of animal wastes produced from slaughtering cause environmental pollution by releasing toxic substances, leachate and greenhouse gases. Direct application of animal carcasses to agricultural fields can adversely affect soil microbiota. Effective waste management technologies such as thermal drying, composting, vermicomposting and anaerobic digestion transform animal wastes, making them suitable for soil application by supplying soil high in organic carbon and total nitrogen. Recent agronomic practices applied recycled animal wastes as organic fertilizer in crop production. However, plants may not survive at a high fertilization rate due to the presence of labile carbon fraction in animal wastes. Therefore, dose calculation and determination of fertilizer application frequency are crucial for agronomists. Long-term animal waste-derived organic supplementation promotes copiotrophic microbial abundance due to enhanced substrate affinity, provides micronutrients to soils and protects crops from soil-borne pathogens owing to formation of plant-beneficial microbial consortia. Animal waste-derived organically fertilized soils possess higher urease and acid phosphatase activities. Furthermore, waste to fertilizer conversion is a low-energy requiring process that promotes circular bio-economy. Thus, considering the promotion of soil fertility, microbial abundance, disease protection and economic considerations application of animal-waste-derived organic fertilizer should be the mainstay for sustainable agriculture.

**Keywords:** animal waste recycling; organic fertilization; agronomic efficiency; soil health; bio-economy; environmental sustainability

## 1. Introduction

Due to increase in the world's population and changes in their dietary habits, the global demand for food is expected to be doubled within the next few decades [1]. Particularly in India and China, the adoption of a more westernized diet can contribute about 50–70% of the total growing need as both countries together represent 37% of the world population [2]. According to Food and Agriculture Organization (FAO) of the United Nations, the world population may reach 9 billion by 2050 [3], which will create huge pressure on our growers to further accelerate agricultural production. This may be achieved either by improving the farming systems or by increasing agricultural land use [4]. To feed the constantly growing population, Sjauw-Koen-Fa [5] estimated 9% expansion of arable land, 14% increase in cropping intensity and 77% more yields, while Pretty and Bharucha [6] suggested sustainable intensification of agro-ecosystems rather than the enhancement of cultivable land. The Green Revolution in the late 1960s aimed to alleviate extreme poverty, malnutrition and hunger of millions. This movement converted farming to an industrial

system, which incorporated the application of modern machinery, the use of synthetic agro-chemicals and genetically modified organisms to the agro-ecosystems [1]. Unfortunately, soil salinization caused by mineral weathering and human interventions and shortage of water has made extensive areas of the total cultivable lands unproductive [7].

Chemical fertilizers sustain short-term productivity of agro-ecosystems, while their indiscriminate use reduces soil fertility, adversely affects enzymatic activity and jeopardizes copiotrophic community [8]. Copiotrophs (i.e., fast growing, *r*-strategist), are a group of microorganisms that thrive in an environment rich in organic matter, particularly carbon. They are nutritionally opposite to oligotrophs (i.e., slow growing, *k*-strategist) that live in much lower C concentrations [9]. Surviving in a nutritionally deprived/rich environment must involve expression of different sets of genes among the copiotrophs and oligotrophs. Koch [10] stated the possible reasons for oligotrophs to succumb during challenges of too high nutrition are: (a) Sudden availability of too many transportable non-metabolic substances, (b) cell death due to osmotic swelling, (c) inappropriate SOS response that means blockage of DNA synthesis and (d) generation of inactive or variable but not cultivable cells. Conventional farming practices and associated synthetic fertilizations not only pollute ground water sources but also put an unbearable burden on our farmers [11]. The majority of these inorganic substances are persistent [12], not readily degraded by natural microorganisms, which can reduce soil viability and negatively affect the quality of produce [13]. Organic agriculture is thus going to be an effective alternative worldwide. Such practices encourage quality food production excluding the use of pesticides and synthetic fertilizers, and targets equilibrium in soil dynamics. The addition of animal-derived amendments to soils promotes plant growth, crop yield and fruit quality due to balanced supply of C, N and P as well as supplying micronutrients such as zinc (Zn), manganese (Mn), boron (B), copper (Cu) and iron (Fe) for crop improvement [14,15]. Organic farming makes the soil friable and fertile: It enhances cohesion of aggregates, accumulates massive organic matter (SOM) in soil and influences microbial communities and their co-occurrences [16]. Moreover, this can improve N use efficiency of crops significantly and reduce $NH_3$ loss through volatilization [17]. Increased SOM supports vigorous uptake of nutrients, provides food for indigenous microorganisms and contributes greater C sequestration in organic agro-ecosystems, as well as having a strong effect on soil aggregation [18,19]. Aggregation size, classes and stability also affect microbial communities of soil and their composition and diversity. For example, alpha-*Proteobacteria* was dominant in macro-aggregates with high SOM. Organic matter is divided into stable and labile fractions. Stable C fractions are highly resistant to microbial decomposition, while labile C proportions have a rapid turnover rate and are directly related to the plant nutrient supply [20]. Labile C fractions are *O*-substituted alkyl, carbonyl and methoxyl C generally abundant in animal waste that are readily available to copiotrophs as energy source [21]. Recently, Bhunia et al. [15] found copiotrophs to be abundant in soils fertilized with recycled slaughterhouse waste, although it could vary with the type of organic substances amended [16]. Simultaneously, organically treated soils attained higher urease, dehydrogenase and acid phosphatase activities, which have a key role in nutrient recycling and decomposition of soil organic matter [22]. Chae et al. [23] considered β-glucosidase as a biological indicator of soil ecosystem health. In addition, organic fertilizers protect crops from *Pythium*, *Fusarium*, *Verticillium*, *Phytopthora* and *Rhizoctonia* like soil-borne pathogens [24], while excessive chemical fertilization posed a greater risk of pest outbreak [13]. Currently, the worldwide crop production is reduced by 36% due to emerging diseases of plants and regular pest attacks [25]. The application of organic manures provides a source of food substrates of varying quality that invoked competitions among microbial communities changing the structure and function of resident soil microbiome [26]. Few studies suggested that the production of volatile and non-volatile toxic compounds released during the decomposition of supplied organic amendment can also be the attributor of plant disease suppression [27,28]. Sturz and Christie [29] documented parasitism, competition, antibiosis, and systematic induced resistance (SIR) as possible mechanisms for allelopathic exclusion of soil-borne

phytopathogens. Organic cultivations may gain fast acceptance globally if adverse effects of inorganic substances are highlighted with fervor.

Over the last decade, livestock production in India has also expanded. Following the slaughtering of animals, meat sectors generate organic wastes in vast quantity, which comprises 45% of the animal body weight [30]. Due to lack of government policies and proper awareness, these are either incinerated or landfilled in developing countries like India. Landfilling and/or open disposal of livestock mortalities can pose a serious threat to environment as they are the latent reservoir of Avian influenza, *Salmonella*, *Bacillus*, *Brucella*, *Clostridium*, *Campylobacter* and bovine spongiform encephalopathy (BSE) [31]. In addition, dumping sites release various toxic compounds, leachate and $CH_4$ and $CO_2$ like major greenhouse gases (GHGs) together with annoyances from bad odors to their surroundings [32]. The unscientific management of such wastes carries calculable risks to human health. For example, zoonoses caused by the infectious livestock pathogens may increase the morbidity of farm workers [33]. Indeed, animal wastes are a rich source of organic nutrients and will propagate environmental pollution if they are not utilized responsibly. Agronomic practices with untreated animal waste can introduce organic pollutants in the agro-environment and increase the number of antibiotic-resistant bacteria in soil [1]. The propagation of antibiotic resistant genes in bacteria is driven by horizontal gene transfer (HGT), and rhizosphere soil has been considered as a major hotspot for HGT [34]. Interestingly, the accumulation of heavy metals in agricultural field due to repetitive overuse of animal-derived amendments and/or raw organic waste may enhance antibiotic resistance in indigenous bacterial population [35]. The evolutionary theory of co-selection, which drives cross- (where same gene provides resistance to both antibiotic and heavy metals) and co-resistance (when resistance is offered by different genes of the same genetic loci) is found most relevant with the occurrence and propagation of antibiotic resistant genes in agricultural soil [1]. Due to emergence of antibiotic-resistant genes, an estimated 10 million human deaths will occur per year by 2050 [36]. Moreover, even plants die due to overuse of raw organic amendments in agriculture [15,21]. To minimize such adverse effects and to ensure biosecurity, waste recycling is necessary before reusing them in agriculture, which may also support the concept of circular bio-economy. Generally, animal wastes are treated through composting, vermicomposting, anaerobic digestion and drying methods. Sometimes, rendered meals are also used in organic farming and aquaculture [37]. Special emphasis has been given to these conversion technologies and their effects on pathogen and heavy metal removal from waste as these are the major challenges in waste to fertilizer conversion.

The emerging problems in developing countries like India appear clear: (a) Production of safe, healthy and affordable food for the constantly growing population, (b) recycling and reuse of organic waste in agriculture as fertilizer, (c) reduction in GHG emissions and environmental pollution, (d) protecting soil health and landscape diversity from synthetic applications and (e) development of a bio-based economy to achieve overall sustainability. To meet the future climatic and socio-economic challenges, agriculture needs to be organic and more productive. Adoption of Organic Agriculture 3.0 that aims to shift organic cultivation from its current domain to mainstream, may solve the problems linked with food safety and environmental health and provide an opportunity for the organic sector development [38]. This version is an advancement over the previous Organic Agriculture 2.0 and 1.0, which included socio-economic empowerment of rural areas, agro-ecological intensification and development in food production incorporating novel ethics and habits [39].

Our recent work, Bhunia et al. [15], demonstrated recycling of rural slaughterhouse waste through tray-drying and showed its agronomic efficiency during successive cultivation of bell pepper and amaranth, where N fertilization at a rate of 80 kg ha$^{-1}$ produced higher yield and better fruit characteristics. Earlier, Roy et al. [40,41] applied sun-dried mixture of bovine blood and rumen digesta as major N source for the cultivation of solanaceous vegetables in India. On the other hand, high-temperature pyrolysis converted bone-based

animal waste to an alternative of rock phosphate for fertilizer production [42]. During pot cultivation of maize, Frazão et al. [43] used granular poultry litter as an effective P substitute. Furthermore, Nunes et al. [44] grew soybean and corn with composted abattoir waste, while Arancon et al. [45] assessed positive effects of animal manure vermicompost. Feasibility of these organic fertilizers varied considerably with the feedstock type and adopted treatment technology [46]. Indeed, crop nutrient-use efficiency is highly dependent on the carbon-to-nitrogen (C/N) ratio of the substances applied during cultivations as lower C/N value indicated a high fertilizer quality and net mineralization of N in soil [47].

Therefore, this review article aims to discuss: (a) Animal waste recycling and their reuse in agriculture, (b) agronomic efficiency of the animal-derived fertilizers and (c) their potential effects on biology and fertility of soil agro-ecosystems as well as (d) to develop a bio-based economy through waste-to-fertilizer conversion.

## 2. Recycling Animal Waste for Fertilizer Production

### 2.1. Composted Amendment

Under the framework of circular bio-economy, waste recycling is currently gaining much interest instead of burial and burning of livestock mortalities. Following the European Union (EU) Directive 2002/1774/EC, such traditional practices are strictly prohibited within the EU [48]. As animal wastes contain high fat and protein, they can be used as viable feedstock in various value-added applications [49]. Indeed, composting is still preferable over the decades for recycling of animal waste in agriculture, which is relatively inexpensive and also environmentally acceptable.

Biological degradation of organic waste is carried out either in presence of oxygen or in anaerobic mode. Composting is an aerobic route of organic waste valorization that typically occurs in four consecutive phases, namely mesophilic, thermophilic, cooling and maturation. During the initial stage of composting, mesophilic activities increase compost temperature up to 68 °C that can facilitate faster proliferation of thermophiles [50]. Therefore, the second phase entails efficient eradication of pathogens to ensure biosafety. However, some opportunistic pathogens such as BSE and *Salmonella* may re-colonize the compost when temperature begins to decrease in subsequent cooling stage [51]. At maturation, the composted material becomes a friable, inodorous and humus-like nutrient rich product that can replace commercial inorganic fertilizers, improving soil properties through supplying essential crop nutrients. The main factors that could affect the quality and content of compost are waste C/N ratio, mode of composting, decomposition conditions and addition of nutrients during the process [46].

Three types of composting systems are currently employed, including windrows, static bins and in-vessel composting. Franke-Whittle and Insam [30] recommended windrow composting of animal carcasses as it reduced the pathogenic load better than the in-vessel type. According to Senesi et al. [52], the presence of humic acid fractions in organic compost makes it ecofriendly and agronomically acceptable. Animal waste composting can be considered beneficial in terms of the microbial stabilization, pathogen inactivation, moisture reduction and good fertilizer quality of the end product [53]. Ragályi and Kádár [54] applied slaughterhouse compost as fertilizer in agriculture. Poultry hatchery waste comprised of infertile eggs, dead chickens, decaying tissues and blood contaminated wastewater which contained 1% N, 2.5% P and 0.2% K when it was composted with poultry litter as reviewed by Glatz et al. [55], while Nunes et al. [44] found 18.2% organic C, 1.8% N and 2% P in cattle manure compost. Furthermore, fertilization with immature compost can increase soil salinity, facilitate N immobilization due to higher C/N ratio and suppress plant growth enhancing osmotic stress [56], whereas maturate compost with a C/N ratio lower than 20 may act as valuable soil conditioner [50]. Moreover, Bhunia et al. [31] suggested additional thermal treatments to make the compost pathogen-free.

### 2.2. Vermicompost Manure

Like composting, this technology is also involved in organic waste stabilization under aerobic environment. Vermicomposting has emerged as a sustainable option with two major benefits of transmuting plant available nutrients into much more soluble forms along with simultaneous reduction in total as well as bioavailable heavy metals content [57]. Simply, vermicompost is a mixture of worm cast, humus, live earthworms and their cocoons. It is a finely divided peat-like material that is rich in NPK, essential micronutrients and beneficial microbial communities including $N_2$-fixing, P-solubilizing bacteria and actinobacteria [58]. Today, vermicompost has become an imperative component of organic farming systems as the product has better nutrient profile and higher microbial status than traditionally available compost manures [59]. In general, vermicompost fertilization improves quality of agricultural produce and can be physical, chemical and biological attributors of the soil health.

Vermicomposting, an efficient biotechnological approach of composting, employs certain earthworm species to turn the organic waste into nutrient-rich manure, during which organic fractions of solid waste are modified by associated microbial communities secreting hydrolytic enzymes, while the earthworms used accelerate the process through substrate aeration, mixing, grinding, fragmentation and enzymatic digestion [57,60]. Out of the thousands of species of earthworms, *Eisenia fetida* was found to be the most appropriate epigeic one for vermicomposting of animal waste and biogas plant slurry [61]. After 10–15 days of primary decomposition, earthworms were incorporated at 8–10 worms $kg^{-1}$ of waste in specified composting bed, where the earthworm activity undergoes two distinct phases: An initial active phase followed by the maturation or aging stage [62,63].

At maturation, vermicompost turns dark-brown, non-sticky and odorless with a final moisture range between 25 and 30% and may then be harvested from the top of the heap separating applied earthworms. Atiyeh et al. [64] cultivated marigolds using pig manure vermicompost. Yadav et al. [61] assessed around 2.8% N, 1% P and 0.9% K from earthworm-processed cow waste, while Garczyńska et al. [65] showed that vermicompost derived from Cameroon sheep dung had an organic C content of 34%, total N of 1.7%, P of 1% and K of 1.3%. Previously, Borges et al. [66] reported that the mixture of cow and swine manure (in 50:50 ratio) provided greater mineral composition in final vermi-produce. The use of immature vermicompost may introduce crop toxicity. On the other hand, disease-causing plant pathogens such as *Pythium*, *Rhizoctonia* and *Verticillium* are suppressed when mature vermicompost is applied at a moderate rate [67]. Moreover, these can incorporate huge amount of organic matter into agricultural soils, thereby improving soil aeration, aggregation stability, water-holding capacity and nutrient availability as well as stimulate enzyme and soil microbial activity. Kumazawa et al. [68] briefly documented beneficial roles of organic matter in sustainable agricultural production.

As vermicomposting technology does not involve any thermophilic phase, complete eradication of livestock pathogens in the final product is not guaranteed, although the process surprisingly reduced enteric virus, fecal coliforms and *Salmonella* strains in various biosolids [69]. Tognetti et al. [59] inoculated earthworms after the thermophilic stage of composting to overcome the drawback. According to Swati and Hait [57], used earthworms can reduce the mobility of metal ions converting them into lesser available forms who also found such mobile metal ions were accumulated in earthworm tissues.

### 2.3. Anaerobically Produced Digestate

Anaerobic digestion is a series of biological process (namely hydrolysis, acidogenesis, acetogenesis and methanogenesis) that facilitates organic matter breakdown in the absence of oxygen to produce biogas that may be an alternative source of energy to replace fossil fuels. A nutrient-rich product is also derived at the end of this process known as digestate. Anaerobic digestion can be either mesophilic (at 35 °C for 15–30 days) or thermophilic (at 55 °C for 12–14 days) and effectively removes pathogens and pollutants from the



digestate produced, which may then be separated into solid and liquid fractions to fertilize agricultural soils [70–72].

A wide array of organic wastes including agrarian, municipal and sewage sludge can be the feedstock of anaerobic digestion [1]. During this process, the plant-available form of N ($NH_4^+$) increases satisfactorily [73]. Möller and Müller [74] reported 2.2% N, 0.4% P and 0.9% K in pig slurry digestate derived from German biogas plants, while some solid digestates contained 51–61% mineral N that suggested their best use would be as fertilizer [72]. Moreover, digestate application could reduce the risks of P runoff as labile-P fractions are significantly decreased in anaerobic digestion [75]. Few studies have addressed the fertilizer value of anaerobically produced digestate. For example, Nkoa et al. [76] found poultry-derived liquid digestate to be more suitable for high N demanding crops with a short-growing period. Loria et al. [77] used swine manure digestate as N source in corn production, while Collins et al. [78] fertilized potato plants with P-rich animal manure digestate. Furthermore, digestates from animal slurries can be an efficient source of nutrients for vegetable production even under the soilless condition as reviewed by Möller and Müller [74].

In recent years, the effects of anaerobic digestion process on pathogen inactivation and pollutants removal have also been considered for sustainable management of soil fertility. The majority of the slaughterhouse pathogens such as *Salmonella*, *Giardia* and *Cryptosporidium* were destroyed just after 30 min of thermophilic digestion [79], whereas Viau and Peccia [80] failed to eradicate such pathogens adopting the mesophilic process. Masse et al. [81] acknowledged composting as a more effective way for reducing antibiotic residues from organic waste instead of anaerobic digestion. Thus, composting was recently performed extensively along with anaerobic digestion to improve the digestate quality [31], while aerobic post-treatment of anaerobically digested poultry waste was suggested by Salminen et al. [82] to reduce its phytotoxic effects.

*2.4. Pyrolysed Biochar*

Biochar is a carbon-rich charcoal-like organic substance obtained from pyrolysis of biomass waste, which is usually applied as soil conditioner/amender in order to improve agro-ecosystem health and crop productivity and can also reduce the adverse effects of phytopathogens [83]. This technology involves thermochemical degradation of waste materials under an oxygen-deprived environment at elevated temperature, and can be divided into three subclasses as conventional, fast and flash pyrolysis depending on their operating conditions. According to Demirbas and Arin [84], a low process temperature and heating rate would maximize the char production, while Uzoma et al. [85] pyrolyzed cow manure at 500 °C temperature to obtain a biochar with 0.1% N, 0.8% P and 3.3% organic C content. Biochar quality parameters namely pH level, surface area, pore structures, functional groups and elemental compositions differ widely with pyrolysis substrate and temperature [86]. Zwetsloot et al. [42] showed how pyrolysis temperature affected availability and chemistry of P in abattoir bone char. In a recent study, Zhang et al. [87] established that higher-temperature pyrolysis reduced environmental risks and heavy metal toxicity in biochar derived from cow manure.

Biochar application may resolve a diversity of issues including site-specific (e.g., reduction in plant available contaminants) to global-scale problems (e.g., atmospheric C sequestration, GHG mitigation) [88]. Due to the presence of recalcitrant C fraction in biochar, such amendments become resistant to microbial attacks and stay in soil for thousands of years, even though high internal surface area and porous structure of biochar facilitate an ideal habitat for colonization, growth and reproduction of bacteria, actinobacteria and mycorrhizal fungi [89,90]. Thus, biochar addition could promote a potential sink for organic C [91]. It also augmented water and nutrient retention, plant growth, enzymatic activity and cation/anion exchange ability of soil as well as prevented surface water eutrophication and environmental deterioration associated with the extensive use of chemical fertilizers [83]. Indeed, biochar can restore phosphorus sustainability in organic

agro-ecosystems. Wang et al. [92] amended soils with poultry-derived biochar rather than raw litter application to reduce the risks of phosphorus leaching, while Glaser and Lehr [93] reviewed that biochars produced from agrarian residues significantly increased P availability in agricultural soils. It was also reported that animal manure biochars contained more organic nutrients compared to the biochars prepared from plant materials [94]. Moreover, biochar usages in combination with other organic amendments keep the soil healthy by positively affecting microbial community structure and soil dehydrogenase activity. Such fertilization favored higher crop yield [95], although the agronomic efficiency of animal-derived biochars was not yet fully explored.

### 2.5. Dried Animal Waste

After proper heat treatment, organic wastes are generally transformed into either animal/ fish feed or nutritious organic fertilizer [49]. Drying is a simple technique of heat and mass transfer that allows low-cost recycling of animal waste in agriculture as stated by Bhunia et al. [96]. EU Directive 1990/667/EC suggested that drying of biomass waste at 133 °C for 20 min may completely eradicate infectious pathogens from the process end product [97]. Kádár [98] utilized dried slaughterhouse compost for sugar beet cultivation in Hungary, while Roy et al. [40] cook-dried the mixture of abattoir-derived bovine blood and rumen digesta (BBRDM) in different ratios (1:1, 2:1 and 3:1) to assess its fertilizer potential, reduce the extensive use of chemical fertilizers and to provide a clean environment around rural slaughterhouses. During field cultivations of tomato, Roy et al. [41] applied the same fertilizer (3:1), which contained 4.9% N, 0.6% P and 0.9% K along with a 4.8 C/N ratio. In our recent study, Bhunia et al. [15] showed agronomic potential of tray-dried slaughterhouse waste, where the mixture (in 3:1 ratio) was dried at 100–120 °C for 6–8 h using a designed tray dryer system. Drying type and process temperature had significant influence on the quality of end product as we observed during our research. We also developed a new drying equipment for on-site production of the fertilizer in India. A patent has been filed on this equipment by Bhowmik et al. [99] with application number 202031033116. On the other hand, Roy et al. [100] showed the effective eradication of *Mycobacterium*, *Salmonella*, *Clostridium*, *Bacillus*, *Brucella* and *E. coli* O157:H7 adopting the drying technology. Our previous study [15] also confirmed the absence of the above-mentioned abattoir pathogens in BBRDM-fertilized soils through 16S rRNA metagenomic study. Table 1 summarizes waste conversion methods and fertilizer quality of the final produce.

**Table 1.** Nutritional status of organic fertilizers derived from different animal sources.

| Amendment Type | Used Feedstock | Fertilizer Value (%) | | | References |
|---|---|---|---|---|---|
| | | N | P | K | |
| Composted fertilizer | Poultry hatchery waste | 1 | 2.5 | 0.2 | Glatz et al. [55] |
| | Cow | 1.8 | 2 | 0.1 | Nunes et al. [44] |
| Vermicompost manure | Cow | 2.8 | 1 | 0.9 | Yadav et al. [61] |
| | Sheep | 1.7 | 1 | 1.3 | Garczyńska et al. [65] |
| Anaerobic digestate | Poultry | 16.4 | 2.4 | 1.9 | Salminen et al. [82] |
| | Pig | 2.2 | 0.4 | 0.9 | Möller and Müller [74] |
| Pyrolysed biochar | Cow | 0.1 | 0.8 | - | Uzoma et al. [85] |
| Dried amendment | Buffalo | 4.9 | 0.6 | 0.9 | Roy et al. [40,41] |

## 3. Dose Calculation and Yield Potential Assessment

Today, the use of fertilizers in agriculture is obvious in order to meet the growing need for food. In general, fertilization maximizes crop productivity and yields better quality of produce by supplying essential plant nutrients directly or indirectly to the soils. Bhunia et al. [15] observed early-stage mortality of bell pepper plants when cultivated with

excessive supply of N through BBRDM fertilization (180 kg N ha$^{-1}$), while N application at low (80 kg N ha$^{-1}$) and moderate (120 kg N ha$^{-1}$) rates showed profound crop yield and better fruit quality. Previously, Roy et al. [40] experienced the same problem when cook-dried abattoir waste was added to soils in higher quantity (180 kg ha$^{-1}$) at the time of planting, where fertilizer N content was not considered for dose calculation. Indeed, the presence of labile C fractions in animal waste and higher accumulation of NH$_4^+$ -N in soil due to their overuse may arrest vegetative growth of plants and induce phytotoxicity [15,21]. Previously, Bonanomi et al. [101] claimed a significant reduction in phytotoxicity, which was associated with the progressive decrease in *O*-alkyl-C fractions. Lazcano et al. [102] found higher tomato plants death due to rigorous use of compost manure and suggested that the application dosages need to be well controlled. Furthermore, Lim et al. [67] reported the application of vermicomposted manure at a higher rate could reduce crop yield due to availability of soluble salts in vermicomposts. In the case of anaerobically produced digestate, some ambiguity also exists over its agronomic effectiveness. Gutser et al. [73] stated that crops, mainly the vegetable varieties, were unable to uptake readily available form of N in a huge amount, which led to greater leaching risk. Therefore, special emphasis should be given on fertilizer dose calculation and its application frequency determination.

According to Jackson and Smith [103], while studying the effects of fertilizer application rate and time on grain yields, noted that animal manures are a potential source of N for cereal crops. Sradnick and Feller [104] established that commercially available organic fertilizers obtained from plant and animal sources generally contained a higher amount of N than P. Moreover, Hua et al. [105] showed that animal manure application increased crop productivity enhancing N use efficiency in the soils of a 40-year soybean–maize rotation. Based on the above discussion, we also realize that agronomic efficiency of animal-derived fertilizers is highly dependent on its N availability. In contrast, various studies considered animal waste as a source of P in sustainable agriculture [42,75,106]. Judicious use of animal manures maximizes economic returns increasing crop yield per unit of fertilizer applied [107]. Agronomic efficiency (AE) is a metric that includes yield potential of an applied fertilizer and relates directly to economic return [108]. In the majority of the reports, agronomic efficiency was calculated straightforward: Yield data/rate of fertilizer application as stated by Vanlauwe et al. [109], although during our study, we have calculated agronomic efficiency adopting the formula of López-Bellido and López-Bellido [110]. Mathematically, this can be expressed as:

$$AE = \frac{(\text{Yield in fertilized soil} - \text{Yield in unfertilized soil})}{\text{Quantity of fertilizer supplied}} \quad (1)$$

During the field cultivation of wheat, Koutroubas et al. [111] found no significant differences in dry matter yield between the control and soils fertilized with 16 t ha$^{-1}$ farmyard manure (FYM), while the application of composted animal manure (composting of farm wastes along with poultry manure in 3:1 ratio) at a rate of 10 t ha$^{-1}$ attained greater maize productivity than 4 t ha$^{-1}$ as reported by Adediran et al. [112]. Roy et al. [41] applied 225 kg ha$^{-1}$ dried mixture of bovine blood and rumen digesta (produced in 3:1 ratio as mentioned earlier) to obtain 33 t ha$^{-1}$ tomato yields. Authors claimed that, during the cultivation, they had provided 68.31 kg N ha$^{-1}$ of soil, whereas Adekiya and Agbede [113] recorded 7.6 t ha$^{-1}$ yield of tomato supplying 30 t ha$^{-1}$ poultry manure (PM). In another study where the effects of poultry manure, wood ash and rice bran were evaluated, Moyin-Jesu [114] found that relative to other treatments, the application of 6 t PM ha$^{-1}$ provided better cabbage head yield. Similarly, Evanylo et al. [115] assessed the effectiveness of commercial fertilizer, poultry litter and compost-based manures in an organic vegetable cropping system who reported the highest maize growth around 16.2 t ha$^{-1}$ in soils fertilized with 2 t ha$^{-1}$ dried poultry litter (DPL) as shown in Table 2. In contrast, PM addition did not affect maize yield satisfactorily as reported by Busari et al. [116]. On the other hand, Ragályi and Kádár [54] preferred fertilization with composted cattle waste (CCW) at a 25–50 t ha$^{-1}$ application rate instead of chemical use for higher triticale production in

Hungary. After three years, Nunes et al. [44] cultivated soybean and maize plants with the same fertilizer at different dosages (0, 4, 8, 12 and 16 t ha$^{-1}$) and found a quadratic relationship between the crop yield and fertilization rate. Furthermore, Das et al. [117] confirmed that composted cattle waste was agronomically more efficient than applied swine manure compost in rice paddy. On the other hand, fertilization with vermicompost manure also demonstrated the same trend of grain yield. For example, Arancon et al. [45] recorded 16 t ha$^{-1}$ marketable yield of bell pepper upon the application of 10 t ha$^{-1}$ vermicomposted cow manure (VCM). In addition, Llaven et al. [118] showed that peppers treated with sheep manure vermicompost produced better-quality fruits, although vermicompost derived from cow manure (CMV) was not satisfactory according to Joshi et al. [119]. However, the market acceptance of vermicomposts is greater than the composted products probably due to the better visual aspects, larger nutrient concentrations and higher microbial population size and activity [59]. According to Rayne and Aula [120], the degree to which manure affects agro-ecosystem potential is not only dependent on its fertilizer value, but also on the rate and timing of application, soil type and climatic conditions. Conversely, the use of anaerobically produced digestate and biochar in organic farming is not very popular. Alburquerque et al. [121] achieved 44 t ha$^{-1}$ crop yield by adding 6 t ha$^{-1}$ digested pig slurry (DPS) to the field soils. Bougnom et al. [122] applied solid anaerobic digestate, rather than the manure, to obtain higher yield of hay plants. Furthermore, Uzoma et al. [85] assessed the yield potential of cow manure biochar (CMB) during maize cultivation in Japan. Moreover, pyrolyzed biochar could reduce the risk of nutrient leaching [92]. According to Karim and Ramasamy [123], lower fertilization rates always tended to higher fertilizer use efficiency, thus agronomic efficiency was also higher. Vanlauwe et al. [109] established a negative exponential relationship between the AE and amount of N fertilizer supplied, while Chuan et al. [124] showed a positive quadratic correlation among yield response and AE for NPK dosages as illustrated in Figure 1. Simply, higher agronomic efficiency indicates more proficient use of nutrients mainly N by the crops, although developing countries generally practiced price-based selection of fertilizer. Table 2 represents the yield potential of various animal-derived amendments at different dosages where the fertilization rates and crop yield are calculated on a dry matter basis.

Lund and Doss [125] found a strong residual impact of dairy cattle manure on plant health and soil fertility. McAndrews et al. [126] evaluated residual effects of composted swine manure measuring growth and yield parameters of field soybean. Authors reported 0.2 to 0.5 t ha$^{-1}$ productivity, which was higher than the control as well as urea-treated residual plots. Ragályi and Kádár [54] provided evidence of greater residual fertility in soils treated with CCW even after 3–4 years of cultivation. Recently, Bhunia et al. [15] proved that dried animal waste was residually more efficient than the soils treated with chemical fertilizers and market available vermicomposts.

**Table 2.** Yield potential of various animal-derived amendments at different dosages (expressed as dry matter basis).

| Study Country | Fertilizer Type | Application Rate (t ha$^{-1}$) | Cultivated Crops | Yield Response (t ha$^{-1}$) | References |
|---|---|---|---|---|---|
| Greece | FYM | 0<br>16<br>32 | Wheat | 3.2<br>3.4<br>4.5 | Koutroubas et al. [111] |
| Nigeria | FYM+PM (3:1) compost | 0<br>2.5<br>5<br>7.5<br>10 | Maize | 1.6<br>2.1<br>2.2<br>2.4<br>4.0 | Adediran et al. [112] |
| Nigeria | PM | 0<br>5<br>10 | Maize | 1.9<br>3.7<br>2.9 | Busari et al. [116] |
| Hungary | CCW | 0<br>25<br>50<br>100<br>200 | Triticale | 5.2<br>5.4<br>4.7<br>6.7<br>6.4 | Ragályi and Kádár [54] |
| India | CMV | 0<br>5<br>10<br>20 | Wheat | 2<br>3<br>3.1<br>3.1 | Joshi et al. [119] |
| United States | DPL | 0<br>2 | Maize | 2.4<br>16.2 | Evanylo et al. [115] |
| Japan | CMB | 0<br>10<br>15<br>20 | Maize | 1.2<br>1.3<br>3.1<br>2.4 | Uzoma et al. [85] |

FYM: Farmyard manure, PM: poultry manure, CCW: composted cattle waste, CMV: cattle manure vermicompost, DPL: dried poultry litter, CMB: cow manure biochar.

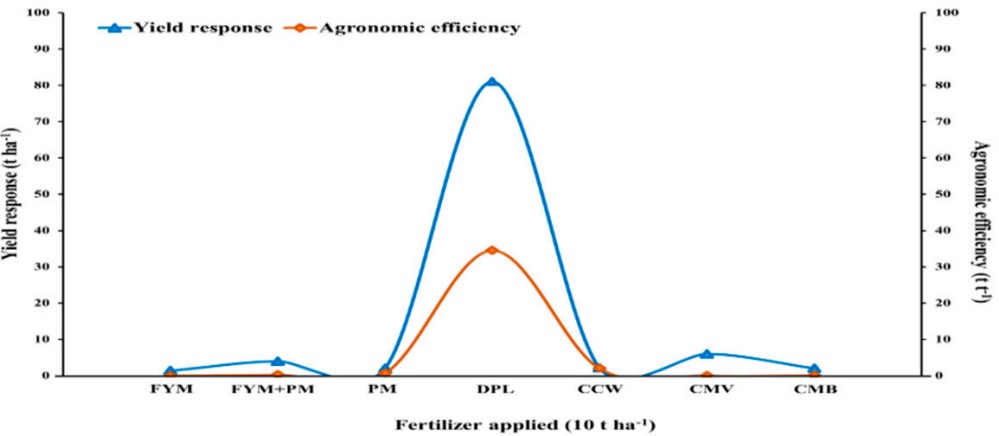

**Figure 1.** Positive quadratic correlation between the yield response and AE, where application rate was kept constant at 10 t ha$^{-1}$ as dry weight basis for all the used fertilizers. This curve behaves quadratically as the highest exponent of the variable in the curve-equation was a square and the relationship found positive as increasing one variable increases the other one. Details provided in Table 2. FYM: Farmyard manure, PM: Poultry manure, DPL: Dried poultry litter, CCW: Composted cattle waste, CMV: Cattle manure vermicompost, CMB: Cow manure biochar.

## 4. Effects on Agro-Ecosystem Health

### 4.1. Aggregate Formation

Organic fertilization through the use of recycled animal waste provides sufficient strength for building up soil fertility especially in regions where soils are nutritionally poor [127]. Healthy soils are by default stable, resilient to stress factors and largely diverse with numerous taxa that form a complex food web through high levels of nutrient recycling [128]. The formation of stable aggregates may sustain crop productivity improving soil structure that provides pathways for the transportation of water, elements and gases as well as facilitates an ideal environment for microbial growth. Interestingly, aggregation stability is mainly dependent on the SOM content and dynamics [129]. Few studies reported that long-term inorganic fertilization promoted cohesion of aggregates [130–132], while no changes or even decrease was observed by Bandyopadhyay et al. [133], Xin et al. [134] and Zhou et al. [135] in comparison to unfertilized plots. In contrast, the addition of organic amendments enhanced SOM bound soil particles together into aggregates. Zhang et al. [136,137] showed a positive correlation between the aggregation stability and associated binding agents. Furthermore, it was reported that organic application decreased the proportion of micro-aggregates (<250 μm) with the mean geometric diameter and accumulated more SOM in macro-aggregates (>250 μm), as shown in Table 3. According to Lin et al. [16], micro-aggregates with recalcitrant SOM had less favorable habitat conditions due to greater cooperation and competition among the microbial groups, and found that the classes *Gaiellales* and *Pezizales* were abundant in micro-aggregates. Indeed, micro-aggregates with lower SOM chose *Actinobacteria* adopting the *k*-selection strategy, while *Proteobacteria* was dominant in macro-aggregates with high labile SOM [138]. In addition, Ma et al. [139] demonstrated that a deficiency in labile SOM enhanced both competition and cooperation among soil microbes. However, the degree of aggregation was highly influenced by the type of organic fertilizers amended and the presence of indigenous microbial communities in arable soil. Guo et al. [140] incorporated straw manure to improve structural stability of the soil, while cattle manure appeared to increase macro-aggregate proportion [141]. Furthermore, Lin et al. [16] showed more effective soil aggregation, around 30.6% larger aggregate formation in soils treated with pig manure instead of plant residues or inorganic NPK, whereas Babalola et al. [142] reported a 15.7% increase in aggregation stability after the addition of green manure compost. In addition, poultry litter usages reduced the formation of micro-aggregates by 34% compared to chemical fertilizer treatment and stimulated glomalin production [143]. On the other hand, Li-Xian et al. [144] mentioned that animal manure application with high salt content degraded soil structure to some extent.

### 4.2. SOM Turnover

Organic matter is a key component of arable soil, which is essential for long-term productivity of an agro-ecosystem as it contains N, P, C and other nutrients indispensable for growing plants and an energy reservoir for soil heterotrophic fauna [19] and has a priming effect on global carbon cycle as well [145]. Increasing SOM level interestingly decreases bulk density, which augments water retention, air exchange capacity and root proliferation [146]. It is well documented that the extensive use of chemical fertilizers deteriorated soil health mainly reducing the SOM content and associated microbial communities. According to Ali et al. [147], the SOM content in arable soils can be lower due to intensive cultivation throughout the year. In order to increase the SOM level, organic farming is preferred as suggested by Liang et al. [22], Maillard and Angers [148] and Wang et al. [149] because organic agriculture could replenish SOM more than that lost. The SOM cycling, formation and decomposition, is mainly mediated by the structure, composition and activity of indigenous microbial communities [150,151]. Later, Tian et al. [152] stated that fertilization deliberately influenced SOM content and quality affecting the composition of microbial communities. However, Kong et al. [153] reported that the turnover rate could also be influenced by the factors like irrigation, crop-rotation, soil environment and climate change.

A large number of field experiments have revealed that the long-term manure application alone or in combination enhanced SOM content and its fractions, while Gregorich et al. [154] did not find any significant alteration in the turnover rate of SOM under continuous chemical supplementation. Additionally, Whalen et al. [155] reported that the soil organic carbon (SOC) and total nitrogen (TN) concentration were increased up to 2.02 and 0.24 t ha$^{-1}$ yr$^{-1}$, respectively, when composted cattle manure was added to farmland soils. Similarly, Brown and Cotton [156] found a three-fold higher SOC level in compost-amended soil, while Majumder et al. [157] observed NPK + FYM fertilized plots had similar labile C pool in comparison to the control. Bouajila and Sanaa [158] showed that matured compost application responded better to SOM fractions than the fresh or immature one due to the presence of higher stable C. Dass et al. [159], Jayakumar et al. [160] and Zhao et al. [161] experienced the same effects with animal manure vermicompost. Previously, Compton and Boone [162] showed how soil N dynamics was affected by the quality and quantity of SOC. According to Lal [163], fertilization affects SOC pools mainly increasing the humification rate. During the organic cultivation of tomato by land application of dried buffalo waste, Roy et al. [41] evidenced a temporary increase in plant-available soil N that may be due to lack of N immobilization, while in applying animal-derived biochar in organically fertilized soils, Plaza et al. [164] documented a drastic reduction in soil C loss through organo-mineral complex formation. Furthermore, Lin et al. [16] obtained highest TN and SOM value in NPK + pig manure soil. Authors amended plant residue with virgin NPK that did not affect SOM level significantly in comparison to control treatment. However, readily available SOC and high TN content allowed faster proliferation of copiotrophs as reported by Zhan et al. [165], which are involved in decomposition of organic matter to supply essential plant nutrients. We can consider them as potential indicators of healthy soil.

### 4.3. Microbial Abundance and Community Composition

Soil microbes are an integral part of agro-ecosystem health, can be classified as bacteria, actinobacteria, cyanobacteria, fungus, mycorrhizae, protozoa, algae, and each of them has a specific function in maintaining soil quality. According to Moeskops et al. [166], these serve residue decomposition, nutrient cycling, N fixation, C sequestration and stable aggregate formation as well as having a major role in soil-borne disease suppression. Moreover, the ability of organisms to degrade SOM depends on their enzyme secretion potential [129]. An active microflora is therefore crucial for sustainable crop production. Xu et al. [167] considered rhizosphere microbes as early warning indicators of soil health as they respond quickly to environmental changes. Indeed, the abundance, structure and activities of such indigenous microbial communities could be greatly influenced by the composition of plant species, agricultural practices and various abiotic factors as stated previously by Yu et al. [168]. Their responses towards diverse fertilization regimes have been well studied by so many authors over the past several years. Geisseler and Scow [169] found a 15.1% increase in microbial biomass production and diversity upon the application of mineral fertilizers compared to non-fertilized plots, but later on Wang et al. [170] observed a dramatic reduction in bacterial richness under the same fertilization regime. In contrast, Roy et al. [40] obtained higher numbers (in terms of cfu mL$^{-1}$) of total bacteria, N fixing *Azotobacter*, P-solubilizing bacteria, cyanobacteria and fungi in soils fertilized dried abattoir waste, whereas rice husk biochar only increased the abundance of genera *Thiobacillus*, *Pseudomonas* and *Flavobacterium* that contributed to P availability in soil [171]. In addition, Gopal et al. [172] confirmed that the populations of *Azotobacter*, *Azospirillum*, *Nitrobacter*, ammonifying bacteria and P-solubilizers were superior in *Eudrilus* sp. composted cow manure. The repetitive overuse of ammoniacal fertilizers significantly reduced the pH level in soil, which is closely associated with decreased microbial diversity and changes in indigenous community composition, while the addition of animal-derived amendments to soils prevented the acidification problem and related effects on soil microbiota as stated by Sun et al. [173]. However, it is very difficult to understand the complex responses of

microbial communities towards organic and conventional farming as both the fertilization regimes have different bacterial and fungal populations (Table 3).

During their studies, Chaudhry et al. [174], Wang et al. [170] and Li et al. [175] demonstrated copiotrophic abundance in soils, which was managed organically, while Li et al. [176] observed a slow recovery of oligotrophs when SOC and TN levels were decreased in arable field with the progression of time. Oligotrophs and copiotrophs are physiological traits and can be distinguished by their growth kinetics and substrate affinity for metabolism. Chen et al. [177] obtained a higher Michaelis–Menten constant for copiotrophs that usually stayed in environments with high nutrient levels and preferentially consumed the labile C pools, while in contrast, oligotrophs exploit a soil that is nutritionally poor with low energy flow but, have higher biomass yield for each unit of substrate consumed [9]. Thus, oligotrophs are less reactive to abrupt resource availability and relatively slow-growing. Studies suggested that the main driving factor behind such shifts in community compositions may be the type of organic C incorporated and not the application of P and N [178–180]. The inability of copiotrophs to grow under nutrient-deprived condition includes possessing a relatively lower affinity for the substrate combined with a lack of adequate regulatory mechanisms for starvation as reported by Koch [10]. By applying the copiotroph-oligotroph concept to soil microorganisms, we can make specific predictions about the ecological attributions of various taxa and understanding of structure and function of resident bacterial communities in better way. Furthermore, Ding et al. [181] specified the roles of pH in shaping bacterial community structure who found *Proteobacteria*, *Acidobacteria* and *Actinobacteria* to be pre-dominant in the combined application of NPK + organic manure. Jones et al. [182] recorded the highest Acidobacterial abundance in chemically fertilized soils with a lower pH level. However, Shanks et al. [183] documented *Bacteroidetes* as the most abundant phyla in soils amended with composted cattle manure, whereas Li et al. [176] did not find any significant change in relative abundance when they compared the compost with control treatment. Moreover, the genus *Thermogemmatispora* (phylum Chloroflexi) was reported as key stone taxa in pig manured soils by Lin et al. [16] who noticed a decrease in its relative abundance under the NPK + pig manure treatment. On the other hand, fertilization with vermicompost manure at 3.75 t ha$^{-1}$ rate diminished the richness of oligotrophic *Actinobacteria*, *Acidobacteria* and *Gemmatimonadetes* [184]. In a recent study, Bhunia et al. [15] obtained copiotrophic *Proteobacteria*, *Planctomycetes*, *Bacteroidetes*, *Chloroflexi* and *Firmicutes* as dominant in soils from bell pepper rhizosphere following the application of recycled slaughterhouse waste, while their richness diminished when treated with N/P/K = 10:26:26 + urea. Likewise, Wu et al. [185] observed that *Nitrosospira* abundance in organically fertilized soils that belonged to the ß-subclass of *Proteobacteria* improved fertilizer N use efficiency during the cultivation of grapes. Compared to bacteria, fungus poses more oligotrophic features as they prefer nutrient-rich environment to grow, therefore, the species of fungal phylum *Basidiomycota* including *Irpex*, *Pycnoporus*, *Trametes*, *Schizophyllum* and *Fomes* predominantly show oligotrophy in organic soils [186]. However, Wang et al. [170] demonstrated more ecologically similar groups in arable soils after the addition of organic fertilizers that indicated less interaction between the microbes.

### 4.4. Enzymatic Activity

Soil enzymes are also crucial for maintaining agro-ecosystem productivity. According to Das and Varma [187], ß-glucosidase, dehydrogenase, phosphatase, urease and invertase are the major enzymes that are generally found abundant in agricultural soils. The majority of these enzymes directly originate from viable microbial cells [188]. Their activities together with microbial biomass C (MBC) express biological status of the soil at a given time, and therefore enzyme levels can be used to determine the degree of alteration in soil structure. Moreover, Shi [189] established a positive correlation between the organic matter turnover and enzymatic activity in agricultural soils, whereas Lupwayi et al. [190] showed how MBC and activities of some of the enzymes (ß-glucosidase, NAGase, acid phosphomonoesterase and arylsulphatase) that mediate major biogeochemical cycles were

affected by manure applications at different dosages. Additionally, soil enzymes are found highly sensitive to pH changes, while different fertilizers responded differently to the soil pH [191], although it is well documented that these enzymatic activities were enhanced when soils were fertilized with organic amendments. For instance, the activity of alkaline phosphomonoesterase in clay loamy soils was increased up to 300 mg $p$-NP kg$^{-1}$ h$^{-1}$ with the addition of swine manure biochar at the rate of 0.5%, which was around 150 mg $p$-nitrophenyl phosphate ($p$-NP) kg$^{-1}$ h$^{-1}$ in control treatment [192]. Lupwayi et al. [190] obtained 1956 pmol methylumbelliferone (MUF) g$^{-1}$ h$^{-1}$ ß-glucosidase activity in soils amended with composted cattle manure, whereas the NPK fertilized plots had 1534 pmol MUF g$^{-1}$ h$^{-1}$ activity of ß-glucosidase.  Antonious et al. [193] monitored soil enzyme activity before and after animal manure application and found an increased urease and invertase activities after incorporation of vermicomposted horse manure to native soils. Furthermore, Panuccio et al. [194] showed a higher dehydrogenase activity (255 µg trifenil tetrazolium formazan or TTF g$^{-1}$ h$^{-1}$) in loamy-sand soil, which was treated with 50% solid digestate produced from anaerobic digestion of animal manure and maize silage mixture, relative to control treatment (70 µg TTF g$^{-1}$ h$^{-1}$). These positive results reflect higher metabolic profile of soil under organic farming systems.

**Table 3.** Alteration in agro-ecosystem health under different fertilization regimes.

| Soil Health Parameters | Type of Fertilizer Applied | | References |
|---|---|---|---|
| | **Chemical** | **Organic** | |
| Aggregate formation | Increases the proportion of micro-aggregates in soil (<250 µm) | Accumulates more SOM in macro-aggregates (>250 µm) | Lin et al. [16] |
| SOM turnover | No significant alteration in SOM turnover rate | More labile SOC pools in organically fertilized soils | Gregorich et al. [154]/Brown and Cotton [156] |
| Microbial abundance | Oligotrophic (*Actinobacteria*, *Acidobacteria* and *Gemmatimonadetes*) | Copiotrophic (*Proteobacteria*, *Bacteroidetes*, *Firmicutes* and *Planctomycetes*) | Bhunia et al. [15] |
| Enzymatic activity | Relatively less but greater than the control treatment | Higher | Lupwayi et al. [190] |
| Disease suppression | Poses a greater risk of pest outbreak | Protects crops from *Pythium*, *Fusarium*, *Verticillium*, *Phytopthora* and *Rhizoctonia* like soil-borne pathogens | Kim et al. [13]/Bailey and Lazarovits [24] |

### 4.5. Disease Suppression

Research shows that the disease suppression ability of a crop is mainly confined to the biological properties of soil [195]. As highlighted in the literature, organic fertilizers can also be beneficial in terms of their disease suppression potential, while extensive and imbalanced supply of virgin nutrients pose a greater risk of pest outbreak decreasing the natural resistance in crops [13]. In recent years, the emergence of novel soil-borne plant pathogens and their resistant behaviors towards various phytochemicals have become a challenge to agricultural biologists. However, crops grown in organically cultivated soils exhibited lower attacks of pests and diseases as stated by Bailey and Lazarovits [24] (Table 3). Among the considered amendment types, composted fertilizers and vermicompost manures are generally applied to manage plant diseases and pests attacks without affecting the environment and human health, although their effectiveness against phytopathogens is attributed to microbial populations and community interactions present within these recycled products [196]. Yatoo et al. [197] claimed that vermicompost had better ability to resist plant diseases in comparison to commercial compost manures. Previously, Manandhar and Yami [198] found vermicompost to possess highest efficiency of disease control when they compared the effect of composted and vermicomposted fertilizers on foot rot

disease of rice caused by *Fusarium moniliforme.* Arancon et al. [199] reported similar results applying vermicompost manure against spider mites attack to tomato seedlings, while wilt disease of tomato plants caused by the infection of *Fusarium oxysporum* was effectively suppressed when soil fertilized with dairy solid-based vermicompost [200]. Likewise, Szczech and Smolinska [201] assured that vermicompost application derived from animal manures prevented the abundance of pathogenic *Phytophthora nicotianae.* Pane et al. [202] recorded better inhibition in mycelial growth of *Pythium*, *Sclerotinia* and *Rhizoctonia* upon the addition of animal manure-based composts. In contrast, Bonanomi et al. [203] reported *Rhizoctonia solani* to thrive on animal-derived amendments rich in sugar-containing labile carbon fractions. In a continuation, Bonanomi et al. [21] demonstrated that organic amendments with high labile C fractions would be conducive to plant damping-off disease in short-term but, became suppressive after 100–300 days of application. Recently, Tao et al. [204] showed that the addition of bio-organic fertilizer shielded plants from pathogen infection increasing synergistic formation of biofilm at the root–microbiome interface, which may act as a plant-beneficial consortium against the soil-borne phytopathogens. An overall depiction of animal waste recycling and their reuse in farming systems to intensify agricultural productivity is reflected in Figure 2.

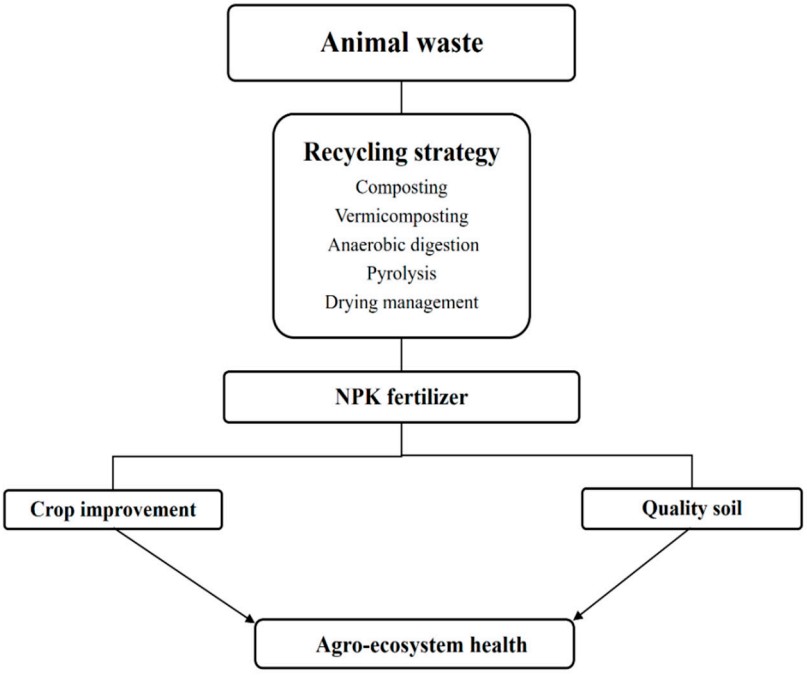

**Figure 2.** An approach to achieve agricultural sustainability through waste to fertilizer conversion.

## 5. Circulation of Nutrients Together with Economy

In this section, we will discuss how a bio-based economy is developed through waste-to-fertilizer conversion. In recent years, the per capita consumption of meat is increasing day after day leading to huge production of animal wastes daily. As we mentioned earlier, most of the time, these wastes are not properly disposed of, which adversely affects the environment and society as well as leading to economic losses. Composting, vermicomposting, anaerobic digestion, pyrolysis and drying are the major treatment alternatives that make such wastes suitable to supply NPK, without which food production would not be possible. This was discussed briefly in Section 2. As animal wastes have greater fertilizer value, as shown in Table 1, these may reduce the use of inorganic NPK sources in agriculture. Waste-to-fertilizer production not only improves the agro-ecology but also can be the backbone of our economy. It is necessary to shift our economic mindset from linear to circular to provide a permanent solution along with societal benefits under the framework of sustainable bio-economy practice. According to the European Com-

mission [205], bio-economy includes the conversion of renewable bio-based wastes into diversified value-added products, which is, by default, circular as described by Carrez and Van Leeuwen [206], Sheridan [207] and Stegmann et al. [208]. This study highlighted a transition from the utilization of virgin nutrients to nutrient cycling, where nutrients are circulated together with economy.

This transition also includes efficient use of nutrients, generates demands for organic fertilizers and measures safe and profitable production and consumption of recycled nutrients as previously recommended by Valve et al. [209]. Figure 3 represents a pictorial overview of circular nutrient economy, which reflects economic, agricultural and environmental sustainability in each step. It starts with the waste production from different livestock farms and meat processing units, and then such wastes are recycled into organic fertilizers like compost, vermicompost, anaerobic digestate and biochar. When these amendments are applied in agriculture, SOM turnover as well as the C/N ratio of soil increases. Such fertilizers also supplies micro-nutrients that are indispensable for plant growth, thus a noticeable improvement in crop productivity was observed by Zhang et al. [14] and Bhunia et al. [15] during their studies. In fact, animal-derived organic amendments can positively affect the structure, nutrient turnover and many other properties of the soil as we briefed in Section 4. Human and livestock consumption of plant produce closed the nutrient loop, which will start again with the slaughtering of livestock animals. This novel approach can recirculate the economy transforming nutrient flows from linear to circular. Adopting this approach, local farmers, livestock owners and meat producing sectors would be benefited creating a ground for profitable business.

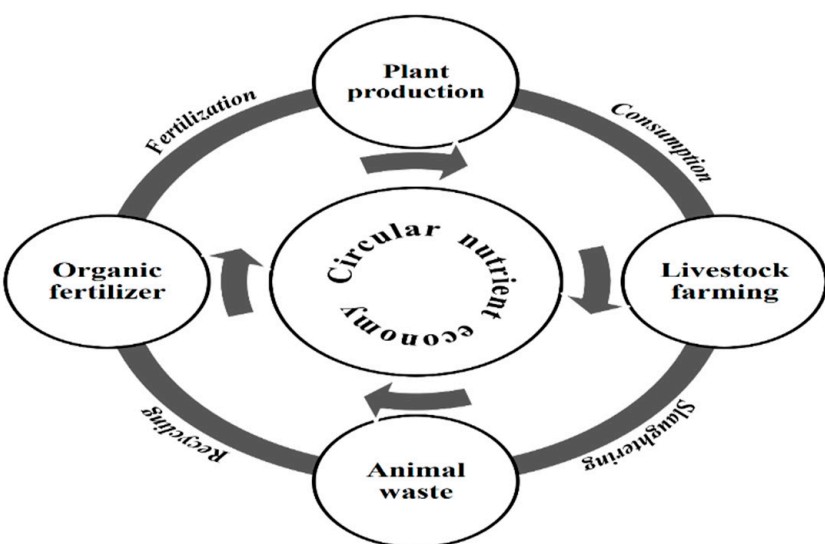

**Figure 3.** Circular nutrient economy and its elements.

## 6. Conclusions

Nutrient recycling through waste to fertilizer conversion safely disposes of livestock waste without polluting the environment. Among the existing conversion alternatives, composting, vermicomposting and thermal drying are relatively cost-effective and environmentally sound methods that can satisfactorily incorporate animal nutrients into the soil agro-ecosystem, which may be lost due to improper disposal of animal waste. No single technology can allow the complete destruction of abattoir pathogens, mainly the re-survival problem of BSE. Thus, the use of techniques in combinations is preferable, although adoption of dry heating technology may be advantageous in rural meat sectors. This review has categorized animal-derived amendments according to their source of origin and treatment technology adopted for recycling and highlighted how dose calculation and determination of fertilizer application frequency are crucial for maximizing crop production and soil fertility. A positive quadratic correlation between the yield

response and agronomic efficiency was established by meta-analysis and the effects of diverse animal-derived fertilizers on soil aggregate formation, SOM turnover, microbial abundance, enzymatic activity and soil-borne disease suppression were also studied. In this study, a special emphasis has been given to the inactivation of waste pathogens that generally contaminate the rhizosphere soil if not treated properly before the land use. We furthermore conclude that, rather than the use of chemical fertilizers, the application of properly recycled animal-derived amendments at the appropriate dose will be more beneficial in terms of cost saving, agro-environmental quality and better crop productivity that should be the mainstay for sustainable agriculture. The main feature of this research is the circular nutrient economy, which exemplified how nutrients circulate together with the economy and affects sustainable development of the society. In view of animal waste valorization, future research on the circular nutrient economy should be encouraged to introduce organic fertilizers into mainstream cultivation.

**Author Contributions:** Conceptualization, J.M.; writing—original draft preparation, S.B. and A.B.; writing—review and editing, J.M.; supervision, J.M. and R.M. All authors have read and agreed to the published version of the manuscript.

**Funding:** This research work received no external funding for publication.

**Data Availability Statement:** No new data were created or analyzed in this study. Data sharing is not applicable to this article.

**Acknowledgments:** Authors thank Anupam Debsarkar, Associate Professor of Department of Civil Engineering, Jadavpur University for his helpful suggestions. We are also grateful to the anonymous Reviewers whose comments vastly improved the manuscript.

**Conflicts of Interest:** Authors declare that they have no conflict of interest.

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
