# Peer review of "Agronomic Efficiency of Animal-Derived Organic Fertilizers and Their Effects on Biology and Fertility of Soil: A Review"

_agronomy, doi:10.3390/agronomy11050823_

Round 1

Reviewer 1 Report

This review aims to make a bibliographic discussion of the information available about the effects of animal-derived organic fertilizers on agronomic efficiency, soil fertility and biology. In fact, the animal waste effect on soil fertility is quite well defined and studied and is not novel. However, the part related to soil microbes and pathogens could be an interesting and important topic, but unfortunately, this part is not well-written and so is not helpful for the scientific community of agronomy and/or soil biochemistry. Generally, this review tried to report the results of others without any explanation about the related mechanisms. How do animal-derived organic fertilizers improve agronomic efficiency, how do they affect soil quality, soil microorganisms, etc? 

Minor considerations:

The abstract is completely confused with short and unrelated sentences. No conclusion. This section needs to be rewritten.

Page 1, line 25:  Such practices protect crops from various soil-borne 25 pathogens. what practices?

Page 2, lines 3-4: Unrelated to the topic. How has the climate helped agricultural development? What is the connection between the previous and next sentence with this sentence?

Page 2, line 9: at the first, you should explain what are the copiotrophic and oligotrophic taxa and what their differences are. Then explain how the bacterial population in rich soil is endangered?! This statement is in stark contrast to the findings reported so far.

Page 2, line 33: How did organic fertilizer protect the plant from pathogens? ‌ By what mechanism? ‌

Page 2, line 53: Why untreated animal waste increased the community of antibiotic-resistance bacteria? Just because of the high level of toxic heavy metal? 

Page 3, lines 20-24: redundant

 Page 3, Lines 39-41: rewrite

Page 4, lines 11-12: rewrite

Page 7 line 5-7: ref? what pathogens? are Bacillus or E. coli considered pathogens?

Page 7, lines 11-26: redundant!

Figure 1: why “quadratic correlation”? based on what values? 

Reviewer 2 Report

Comments to authors

This review explains how to use organic waste. However, there are some parts that are not written correctly and some parts that are difficult to understand.

P1-L15 labile C fraction                                                 I can’t understand this word.

L30-31Recycling of animal …… less energy.             How do you estimate?

P2-L3-4 Climate stability …..to be successful.             Please describe when the climate was stable.

P5-L-23-24 Authors also …. worm tissues.                  Please write a reference.

P6-L42-43 This is a simple ….. in agriculture               I can’t understand this sentence.

P6 [Hatchery] in Table 1.                                                Please explain the materials.

P7-L18-L22 Both nitrate …to a poor root-system.        Please write a reference.

P7-L24-L26 Nitrogen in plants …have too much.        I can’t understand this sentence.

P8-L47-48 We found …… vermicompost addition.     Please write a reference.

P10 Figure 1                                                                   The font size of the figure is small and

                                                                                       difficult to see. Please change.

P10-L20-P11-L2 According … groups.                        Please explain more details.

P12-L27-30 Furthermore, .. …fertilizers,                      I can’t understand this sentence.

P12-L33-36 During their ….over the time.                    I can’t understand this sentence.

P13-L19 p-NP                                                                 Please explain this word.

P13-L47-49 Manandhar ….of disease control.              I can’t understand this sentence.

P15-L26-L27 When applied ……soil increases.           I can’t understand this sentence.

P16 Figure 3.                                                                 I can’t understand this Figure’s meaning.

                                                                                       Please explain the direction between thema.

P-16-L4-7 Waste to fertilizer ….the society.                 I can’t understand this sentence.

P16-L10-14 Nutrient recovery…. supply chain.            I can’t understand this sentence.

Your conclusion or other parts is in-adequate. Please explain about [Recycling of animal waste in agriculture may be an effective 30 alternative to the conventional systems as it consumes about 40-60% less energy.] in Abstract.

Round 2

Reviewer 1 Report

The authors have revised and improved the manuscript and it is now suitable for publication. 

Author Response

Authors thank the Reviewer for spending his/her time on our manuscript and for appreciating our efforts.

Reviewer 2 Report

Comments to authors

I have a large question about Table 2 and Figure 1 of this review.

Is the application rate and yield in Table 2 per fresh or dry matter? If it is per fresh product, it is not theoretical to compare the experimental results with various composts because the water content is very different. If you want to compare the application dose and the yield, I think you should compare per dry matter basis or per total nitrogen basis.

I recommend that you create Table 2 and Figure 1 according to my comments.
